# Relevant Features of Polypharmacologic Human-Target Antimicrobials Discovered by Machine-Learning Techniques

**DOI:** 10.3390/ph13090204

**Published:** 2020-08-21

**Authors:** Rodrigo A. Nava Lara, Jesús A. Beltrán, Carlos A. Brizuela, Gabriel Del Rio

**Affiliations:** 1Department of Biochemistry and Structural Biology, Instituto de Fisiologia Celular, UNAM, Mexico City 04510, Mexico; Rodrigo_Andres1993@hotmail.com; 2Department of Computer Science, CICESE Research Center, Ensenada 22860, Mexico; armando.3eltran@gmail.com (J.A.B.); cbrizuel@cicese.edu.mx (C.A.B.)

**Keywords:** polypharmacological compounds, heterologous machine learning, broad-spectrum antibiotics, human metabolites

## Abstract

Polypharmacologic human-targeted antimicrobials (polyHAM) are potentially useful in the treatment of complex human diseases where the microbiome is important (e.g., diabetes, hypertension). We previously reported a machine-learning approach to identify polyHAM from FDA-approved human targeted drugs using a heterologous approach (training with peptides and non-peptide compounds). Here we discover that polyHAM are more likely to be found among antimicrobials displaying a broad-spectrum antibiotic activity and that topological, but not chemical features, are most informative to classify this activity. A heterologous machine-learning approach was trained with broad-spectrum antimicrobials and tested with human metabolites; these metabolites were labeled as antimicrobials or non-antimicrobials based on a naïve text-mining approach. Human metabolites are not commonly recognized as antimicrobials yet circulate in the human body where microbes are found and our heterologous model was able to classify those with antimicrobial activity. These results provide the basis to develop applications aimed to design human diets that purposely alter metabolic compounds proportions as a way to control human microbiome.

## 1. Introduction

Drug discovery nowadays involves high-throughput screenings of compounds with or without knowledge of the molecular target to treat a condition or disease [1,2,3]; after conducting in vitro and in vivo experiments (e.g., toxicity and efficiency tests), drugs are ready to be tested on humans. It has been noted that although the mechanism of action for some drugs is well studied (e.g., inhibitors of GPCRs [4], protein kinases [5] and penicillin-binding proteins [6]) for many others this is not the case [7]. In fact, it has been recognized that the pharmacological activity of many FDA-approved drugs depends on having multiple targets [8]. This situation has led to drug re-purposing (e.g., viagra [9], aspirin [10]).

In parallel with drug discovery efforts, researchers in biomedicine have discovered the prevalent role of the gut microbiome in human health [11,12]. For instance, the unsupervised antibiotic consumption can induce dysbiosis in gut microbiome that has been associated with celiac disease [13,14] or inflammatory bowel disease [15,16], among others [17].

Considering these two scenarios, polypharmacologic drugs and human microbiome, it has been argued that some FDA-approved human-targeted drugs (FHD) may act through a secondary antimicrobial activity [18]; that is, since the microbiome control different aspects of human health, some drugs that are acting as their primary target on a human protein may also have an antimicrobial activity as a secondary activity. To support these observations, a high-throughput drug screening on human gut microbes was performed using FHD by Maiers and collaborators [19]; 24% of FHD had a secondary antimicrobial activity. The authors noted that antimetabolites and antipsychotics were enriched in 24% of antimicrobial FHD, and found few previous reports about the antibacterial activity on a particular class of antipsychotics. While Maiers and collaborators were concerned about resistance gained by infectious agents exposed to these polypharmacological compounds, here we focus on a different aspect not explored before, the use of antimicrobials as a source to identify polypharmacological compounds. We will focus in this study to polypharmacological compounds that act on human and on microbial molecular targets, which, from now on, we will refer to as polypharmacologic Human-targeted AntiMicrobials or polyHAMWe have previously reported that using peptide and non-peptide antimicrobial compounds (heterologous training set) was an effective method to identify polyHAM from FDA human-targeted drugs [20]. Here we show that polyHAM are more likely found among antimicrobials presenting antibiotic activity against multiple bacterial strains than from antimicrobials acting against a single microbe. Since metabolism is controlled by and controls the microbiome, we tested this model in identifying human metabolites with antimicrobial activity with reliable results. Thus, we report features relevant for the activity of polyHAM compounds that are also found among human metabolites. The implications about these findings in the diagnosis and/or possible treatment of complex diseases are discussed.

## 2. Results

We built three different training datasets: anti-infective, anti-gut1 and anti-gut4 (see Methods). All these sets include the 1181 FDA-approved compounds previously tested by Maier and collaborators for gut antimicrobial activity; consequently, some compounds will change their classification as antimicrobial or non-antimicrobials in these groups as shown in Figure 1. Figure 1A shows the 1096 non-antimicrobials annotated in the three groups, while Figure 1B shows 436 antimicrobials annotated in those same three groups, indicating there are 351 compounds (1532–1181) that are interchanged between antimicrobials and non-antimicrobials in these three groups. The anti-infective set includes 137 compounds annotated as antimicrobials (see Appendix A) by the Anatomical Therapeutic Chemical (ATC) classification [21]; the anti-gut1 set includes 398 compounds (see Appendix A) found to have antimicrobial activity against one or more gut bacterial strains; the anti-gut4 set includes 255 compounds (see Appendix A) with antimicrobial activity against four or more gut bacterial strains; these 255 compounds are also antimicrobials of the anti-gut1 set. In Supplementary Files S1–S3, antimicrobials are instances of class 1 and non-antimicrobials of class 0. The anti-infective is the control group by not including antimicrobial compounds with a known human target so far, while anti-gut1 and anti-gut4 include multifunctional compounds, that is, antimicrobials with a human target. We expect that the number of antimicrobials with human targets would increase from anti-gut1 to anti-gut4, if broad antimicrobial activity would imply acting on multiple targets. Anti-gut1 or anti-infective, on the other hand, are sets with less broad antimicrobial activity than those found in anti-gut4 set. These three sets were used along this study to either: (i) identify all publications on PubMed for those compounds with antimicrobial ontological annotations and, (ii) characterize and classify antimicrobials from non-antimicrobials using molecular features (see Methods).

### 2.1. Publications about Antimicrobial Activity for polyHAM

To identify how many of the compounds with a human target studied by Maier and collaborators had previous publications as antimicrobials, we designed a naive text-mining procedure to identify publications reporting antimicrobial activity for a chemical compound (see Methods). This identification was based on ontological terms associated with broad antimicrobial activity (see Table 1); these ontological terms were identified from each PubMed entry in MedLine format (see Methods). The proportion of publications including antimicrobial terms is presented in Figure 2A–C; this proportion is derived by dividing the number of publications with antimicrobial-related ontological terms by the total number of publications reported (see Methods) for every FHD studied by Maier and collaborators. These results indicate that the broader the antimicrobial activity the more frequent antimicrobial publications: the proportion of antimicrobial publications for dataset anti-gut4 is larger than for anti-gut1 as expected (compare the mean values presented by the black line crossing the boxplots in Figure 2B,C). Although the antimicrobials in the anti-infective set have more publications with ontological terms related to antimicrobial activity than the other training sets, several compounds in the anti-gut4 have more publications with ontological terms related to antimicrobial activity than those found in the anti-infective set, confirming that anti-gut4 set includes compounds with more studied antimicrobial activity than the ones included in the anti-infective or anti-gut1 sets, as expected.

The actual data summarized in Figure 2 are in supplementary Appendix A. Appendix A includes the 16 compounds (out of 203 reported by Maier and collaborators as FHD) that are known to act through a human target having two or more publications with ontological terms associated to broad antimicrobial activity. Our results indicate that the use of ontological terms related to antimicrobials did not identify every polyHAM analyzed. While it is possible to use other terms to identify antimicrobial compounds, our aim here was to explore the broad-spectrum activity of the antimicrobial activity.

### 2.2. Classifying Antimicrobials Using Physicochemical Features

We have previously described a procedure to increase the training set size for antimicrobial compound classification and consequently increase the reliability of predictions; we referred to this procedure as heterologous machine learning, because sets of peptides and non-peptide chemical compounds are used to train models that efficiently classified antimicrobial from non-antimicrobial compounds [20]. Here we further explored this approach to classify antimicrobial compounds in the anti-infective, anti-gut1 and anti-gut4 sets; for that end, we added 7999 antimicrobial peptides and 3546 non-antimicrobial peptides to each of these sets for the heterologous training set construction (see Methods). Our aim is to compare the classification efficiency of antimicrobial compounds using molecular features relevant for heterologous machine learning with that achieved using ontological terms.

To achieve this goal, we characterized the applicability domain of any model derived from these datasets using two general aspects of peptides and non-peptidic chemical compounds. Figure 3 shows the comparison in molecular weight observed between these two sets; as expected, peptides tend to be larger than non-peptidic chemical compounds, ranging from 100 up to 6600 Daltons, with peaks at 300–400, 3000, and 4500 Daltons.

Identifying chemical functional groups from a chemical formula is not a standard or a trivial matter [22]. Indeed, it has been recently noted that no automatic procedure to accomplish this goal leading to the development of a machine-learning-based approach for that goal (see Methods); this machine learning implementation, however, only detects chemical groups that are too broad. For instance, we identified 274 chemical groups in both peptides and non-peptides (see Methods) and observed some similarities in the frequency observed of some of these chemical groups (oxygen, nitrogen, nitrogen aromatic, sulfur, acid, amide; see Figure 4). These chemical groups are different from the features calculated by PaDel-descriptor [23]. To compare the frequency of occurrence of the detected functional group, percentages reported in Figure 5 are normalized per total number of peptides or total number of non-peptidic chemical compounds.

In our previous work, training for drugs that acted against a single gut bacterium identified broad-spectrum antibiotics among FDA approved antibiotics [20]. Our heterologous model in that previous work was not trained to identify broad-spectrum antibiotics, but we were able to identify them because FDA-approved antibiotics are targeted against pathogens while our model would predict these to also act on the healthy gut microbiome. In our current work, we trained our models to predict compounds acting against multiple microbes; hence these models were trained to identify broad-spectrum antibiotics. The best features, model and corresponding parameters were identified using an automatic optimization procedure (see Methods); then the models were tested using a 10-fold cross-validation (see Methods). The use of cross-validation is useful in cases where there is no evidence of noise or incorrect labeling of instances; hence it is a convenient option for our approach.

Twelve different models were generated for each training set: four sets for heterologous training, four sets using only non-peptidic chemical compounds and other four sets using only peptidic chemical compounds (see Methods and Supplementary Files S4–S39). For instance, a set of compounds (e.g., anti-gut1) is complemented with antimicrobial and non-antimicrobial peptides to create a heterologous training set; the original set includes only non-peptide compounds hence are considered as training set with non-peptidic compounds; for the only peptidic chemical compounds, we used the set of antimicrobial and non-antimicrobial peptides used to create the heterologous set. For each of these sets (heterologous, only peptides, non-peptidic chemical compounds), four different representations of the PadelDescriptor features were generated: (a) nominal (classes –antimicrobial and non-antimicrobials, were labeled as nominal), (b) nominal and normalized (nominal plus PadelDescriptor features were normalized and nominal), (c) nominal, normalized and attribute selection (as b plus PadelDescriptor features were selected by the CfsSubsetEval Weka filter), and (d) nominal and attribute selection (see Methods). Figure 5A–C show the true positive rates (see Methods) for antimicrobials and non-antimicrobials achieved by these models on each training set. Noticeable, models trained with anti-gut4 achieved the best classification rates and heterologous datasets (square symbols in Figure 5) rendered some of the best models followed by models trained with only peptides (triangles in Figure 5); these results indicate that polyHAM are to be found among antibiotics that act on multiple species, as expected. The confusion matrices and MCC score for every model are reported in Supplementary Files S40–S42; the best models achieved MCC scores above 0.95.

The names of the models found for all training sets are presented in Appendix A. The best model (dataset anti-gut4 using heterologous and nominal data using the random forest model) was able to classify correctly 89% of every antimicrobial and non-antimicrobial compound in the anti-gut4 set as reported by AutoWeka. The top 10 features used by the best model, out of the 507 features used by the model, are shown in Table 2. Please note that all these features are related to the information or graph theory parameters of chemical groups, rather than chemical attributes; this is in agreement with our previous observation that peptides and non-peptide compounds shared few and too general chemical groups, hence chemical features were not useful for classification purposes. TP rates were used to evaluate the possible bias in classification induced by the biased composition in the training sets. For instance, the anti-infective data set has 137 positive and 1044 negative instances (see Figure 2); yet, we observed that the best models (heterologous sets that added 7999 antimicrobial peptides and 3546 non-antimicrobial peptides, rendering a total of 8136 positives and 4590 negatives) did not favor the classification of antimicrobials or non-antimicrobials as it can be observed in Figure 5A, where the square blue symbols show around 0.9 of TP rate for non-antimicrobial and around 0.7 of TP rate for antimicrobial, a strong influence of the class imbalance should have produced a larger than the observed gap (0.9 vs 0.7) between these two cases (8136 vs 4590).

Based on this data, we performed a test of the 12 anti-gut4 models (four models with heterologous training set, four models with only peptides, and four models with non-peptidic compounds; the four models correspond to four different ways to process the data, see Methods) with a set of 17 metabolites (see Appendix A) that are part of the Human Metabolome Database, which includes metabolites found in the human body. Please note that none of the compounds in the testing set were part of the original training set. While plants and microbes are known to produce secondary metabolites with antimicrobial activity, human metabolites are not commonly recognized to harbor this antimicrobial activity. Applying our naïve text-mining approach, six metabolites on this test set were identified as antimicrobials based on publications that included the ontological terms associated to antimicrobial activity (see Section 2.1) and for the other 11 there was no publication about their antimicrobial activity. According to the ZINC database, the 17 human metabolites have a human target and six have antimicrobial activity corresponding with true polyHAM compounds; four out of these six true polyHAM are broad-spectrum antibiotics (see Appendix A). We observed that in this test set, the best model is derived from non-peptidic compounds as training set using nominal representation of the data (see red circle on Figure 6). We also noted that overall heterologous models classified better the non-antimicrobials than the models trained with only peptides or non-peptidic chemical compounds (see Figure 6): three out of the four models trained with only peptides predicted none of the non-antimicrobials (observation derived from data presented in Figure 6; these three models are observed as a single remarked triangle at the upper left corner of the plot); three out of the four models trained with non-peptidic chemical compounds predicted every non-antimicrobial, but no antimicrobial (observation derived from data presented in Figure 6; these three models are observed as a single remarked circle at the bottom right corner of the plot). Hence in general, the models obtained with heterologous sets rendered more balanced predictions. Yet, it is clear that this test set was not an easy task for any of the models.

Considering the relatively low performance to classify polyHAM in human metabolites, we evaluated how different were the testing set compounds from those found in the training sets. The dataset rendering the best model (anti-gut4 using heterologous and nominal modeling of data) was used to compute the shortest Manhattan distance between the polyHAM in the training and the testing sets (D1); as a reference, we also computed the shortest distance between the non-polyHAM and polyHAM in the same training set (D2); the Manhattan distance was calculated from the vector representation of every compound, in which the vector values were the features calculated for each compound. If D1 > D2 then, the compounds in the testing set were more different to the polyHAM than the non-polyHAM compounds, yet D1 = D2 = 0; as noted in Figure 3 and Figure 4, the diversity in mass and chemical groups in the training set was large, hence this result is in agreement with these previous observations. We then identified the shortest distance between the polyHAM in the testing sets with those of the non-polyHAM in the training set (D3 = 6708088) and compared it with the shortest distance between polyHAM and non-polyHAM in the training set (D2 = 0) and observed that the testing set was more distant from the non-polyHAM compounds in the training sets than the polyHAM compounds in the training set. These results indicate that the testing set was more distant from the non-polyHAM compounds yet, close to the polyHAM compounds in the training set. This distinct distribution of the testing set may provide an explanation for the difficulty in classification for the best models, that is, the best models may have traced frontiers between the positive and negative antimicrobial compounds that excluded several of the compounds in the test set that were too distant from the true positive polyHAM in the training set.

## 3. Discussion

The story of penicillin set a hallmark for antibiotic discovery: it supported the magic-bullet idea proposed by Paul Erlich in which a single drug would traverse the body and only act on a specific target [24]. However, the discovery of antimicrobial peptides as part of the defense mechanism of every cell brought a new concept into the antibiotics field: natural antimicrobial peptides are less likely to evoke resistance in microbes because they act on multiple targets [25]. Thus, while it is possible to kill microbes targeting essential cellular functions coded into a single protein (*e.g*., penicillin-binding proteins), it is also possible to kill microbes by targeting multiple targets. This last view is important for many complex diseases humans are facing nowadays that are related to gut microbiome, such as obesity [26], hypertension [27], among others [28,29], where the antimicrobial activity in drugs is relevant to either treat the disease (by killing microbes associated with the disease) or prevent it (by not killing microbes that prevent the development of the disease).

Drug repositioning or repurposing frequently identifies antibiotics to treat diseases other than infections [30]; such is the case of minocycline that is a semisynthetic tetracycline-derived antibiotic that has shown to have neuroprotective activity and it is currently being tested in treating Parkinson’s disease [31]. Hence, it seems that polypharmacologic antimicrobials that act on human targets, polyHAM, represent a resource to identify effective drugs to treat different conditions beyond infections. In our previous work, we were interested in testing the reliability of heterologous training sets in identifying polyHAM [20]. In this study we characterize several features of these compounds using different computational approaches. We show that there are very few chemical similarities between peptides and non-peptide human-targeted drugs, and consistent with these findings we observed that topological features of chemical structures are more informative than chemical descriptors of molecules for the classification of polyHAM. Thus, for polyHAM to act on multiple targets it is relevant to display specific topological features rather than particular chemical groups. This suggests that the structural organization of chemical groups rather than the chemical groups, per se, are relevant for acting on multiple targets. These results indicate that antimicrobial classifications based on chemical descriptors (see, for instance, [32,33,34]) may not work properly to classify polyHAM.

We used three different training sets for identifying the best model to classify polyHAM: anti-infective, anti-gut1, and anti-gut4; these last two sets differ in the number of bacterial strains that these compounds act against to, increasing from anti-gut1 to anti-gut4, and expecting that anti-gut4 had the most broad ability to act as antimicrobials than the other compounds in anti-infective and anti-gut1 sets. Indeed, we verified by an automatic bibliographic search that anti-gut4 had a larger proportion of known broad antimicrobial compounds than anti-gut1. As noted above, anti-gut4 also rendered the best model to classify polyHAM from non-polyHAM. This result indicates that broad-spectrum antimicrobials make better polyHAM than specific ones and that broad-spectrum antimicrobials are more likely to act on multiple targets.

Finally, human metabolites were used to test our trained model to classify polyHAM; these compounds are not commonly regarded as a source of antimicrobial compounds. Here we show that there is a group of these human metabolites with previous reports about their broad-spectrum antimicrobial activity, hence, these may represent a natural way for humans to control microbiome composition. For instance, the presence of 3-phenylpropionic acid has been shown to be affected by antibiotics altering the healthy microbiome composition [35], and at the same time has been shown to prevent the growth of the pathogenic *Listeria monocytogenes* in combination with a natural antimicrobial peptide [36]. In this case, the quantification of this metabolite may indicate the susceptibility of a healthy individual to be infected by pathogens such as *L. monocytogenes*. Thus, predictions of polyHAM among human metabolites may promote the development of tools to design human diets aimed to alter the specific composition of human metabolites. It is expected that certain diets would promote the accumulation of metabolites with broad antimicrobial activity that, in turn, promote gut microbiome dysbiosis. Further studies are required to validate this idea, yet our findings represent an important advance in that direction.

## 4. Materials and Methods

### 4.1. Dataset Preparation

Three datasets were constructed for the purpose of training machine-learning models, including anti-infective (Appendix A), anti-gut1 (Appendix A), and anti-gut4 (Appendix A); all these datasets were derived from the work reported by Maier and collaborators that included 1181 FDA-approved compounds [19]. These sets differ in the labels identifying compounds as antimicrobials or non-antimicrobials; for instance, in the anti-infective set the compounds were labeled as antimicrobials if they belong to class J in the Anatomical Therapeutic Chemical (ATC) Classification System, in the anti-gut1 set, compounds were labeled as antimicrobials when Maier and collaborators identified antimicrobial activity against at least one gut bacteria and anti-gut4 those hitting at least four gut bacteria. The features were normalized and/or performed a selection of features using the weka.attributeSelection.CfsSubsetEval filter [37].

For each of these three training sets (anti-infective, anti-gut1 and anti-gut4), 12 different sets were generated: four for heterologous data, four for non-peptidic compounds, and four for only peptidic compounds; each of these four sets corresponds with nominal, nominal-normalized, nominal-normalized-selected attributes, and nominal selected attributes, in a similar fashion as described previously [20]. For the non-peptidic compounds set, 1181 non-peptidic compounds were used (the number of antimicrobials and non-antimicrobials depended on the training set, see Figure 7); the only peptidic compounds set included 11,545 peptides (7999 antimicrobial and 3546 non-antimicrobials) that were obtained from a non-redundant compilation of multiple antimicrobial peptide databases [38]; the heterologous set included all non-peptidic (1181 compounds) and peptidic compounds (11,545 peptides). In order to obtain the features describing each compound the PaDelDescriptor software was used [23]; to do that, non-peptidic compounds were represented in SMILES format while for peptides, since the FASTA format is not readable by PaDelDescriptor, they were converted to MOL2 format (version V200) by the program Seq2Mol.jar (see supplementary material to get the code and instructions to execute it). Every feature with values equals to 0 or null in 50% or more of all instances was removed. Finally, 12 different models were generated for each training set: four sets for heterologous training, four sets using only non-peptidic chemical compounds and other four sets using only peptidic chemical compounds (see Figure 7). These training sets were converted into ARFF format (see supplementary files S4–S39). The reliability of every Weka classifier trained with any of these training sets was tested using a 10-fold cross-validation. The script to run the cross-validation test that includes the model name and its corresponding parameters are available at the supplementary Files S43 (anti-infective), S44 (anti-gut1), and S45 (anti-gut4).

### 4.2. Naïve Text Mining Approach

The bibliographic data was obtained from PubChem [39]. The PubChem ID from every compound used in the training sets was obtained from its SMILES formula; from this PubChem ID, every associated PMID was retrieved using PubChem REST server. The PMID was used to retrieve the MedLine format of each publication from PubMed server and to identify the corresponding ontologic terms (entries identified by the headers MH or OT in the MedLine registry). The code used for this purpose is available as a supplementary file named GetPubMedEntriesFromSMILES.java. Figure 3 displays the proportion of references that included an antimicrobial-related ontological term according to the formula:Proportion = NCAMP/TotalNC(1)
where Proportion is the proportion of antimicrobial citations; NCAMP is the number of publications of the compound of interest including an antimicrobial-related ontological term; TotalNC is the total number of publications for the compound of interest.

The test data set was obtained from the ZINC database (version 15) [40] and corresponds with metabolites identified from The Human Metabolome Database [41]. We chose human metabolites for the test set because of the known relationship between microbiome and human metabolism relevant for human health and disease [42]. Every feature describing a compound was calculated as in the training sets, using the PaDelDescriptor software [23]. Briefly, the chemical features correspond with any of the 1444 two-dimensional features calculable by PaDelDescriptor that include chemical features (e.g., acidic groups count, longest aliphatic chain, Hbond donor count) and topological features (e.g., atom type electrotopological state, path counts, topological distance matrix); these features were obtained directly from the SMILES representation of each of the 17 metabolites in the test set. These 17 metabolites were those found with at least one publication including the ontological terms associated with antimicrobial activity identified in the training set. The same features obtained for every model during the training (12 models for anti-gut4) were included for the test set. Antimicrobials were considered those compounds that contained two or more publications with ontological terms related to antimicrobial activity (see Appendix A); the annotation as antimicrobial was done after a human reviewed the literature to confirm the antimicrobial activity.

### 4.3. Machine-Learning Approach

Weka version 3.8 [43] and the AutoWeka [44] plugin were used to train and randomly cross-validate the models. It is worth to mention that AutoWeka includes the state-of-the-art machine-learning algorithms, like SVM, random forest, and logistic regression, among others, so the resulting learning model can be considered as the most suitable for the classification task at hand. For identifying chemical groups (these are different than the features calculated by PaDel descriptor) in peptides and non-peptidic chemical compounds, we used a Python implementation developed for that purpose [22]. Briefly, molecules in SMILES format were integrated into the python code, and the program named rdkitpy was installed as instructed by the developers of Python program; the code searches for 3080 known chemical groups in molecules. True positive rates (TP rates) were used to estimate the proportion of correctly classified instances for antimicrobials and non-antimicrobials; this would allow evaluating for any possible bias in the classification. True positives refer to those instances that were predicted correctly within a class; e.g., an antimicrobial compound that was predicted as antimicrobial is a true positive. Attribute ranking for the best model identified in the training and cross-validation test was performed using the information gain for attribute evaluation filter implemented in Weka; briefly, the information gain for each attribute is derived from the information entropy for each attribute for each class.

All data required to reproduce and analyze the results presented in this work is available through GitHub: https://gdelrioifc.github.io/PolyHAM/.

## 5. Conclusions

In summary, we characterize human-targeted antimicrobials using heterologous training sets and machine learning approaches. PolyHAM display broad-spectrum antibiotic activity and are found circulating the human body where microbes are found.

## Figures and Tables

**Figure 1 pharmaceuticals-13-00204-f001:**
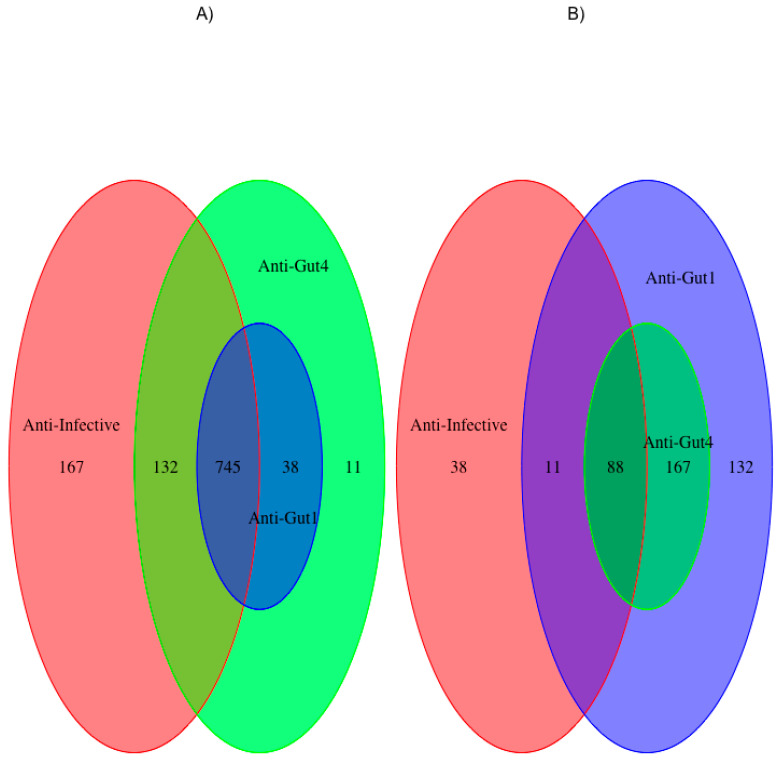
Datasets. The three datasets used throughout this work, including 1181 FDA-approved drugs. Anti-infective set (red ovals) includes 137 compounds that had been classified by the ATC as anti-infective, anti-gut1 set (blue ovals) includes 398 compounds with antimicrobial activity against 1 or more gut bacterial strains, anti-gut4 set (green ovals) includes 255 compounds with antimicrobial activity against four or more gut bacterial strains; the actual compound names and chemical formulas are reported in Supplementary Files S1–S3, respectively. These 3 sets are derived from the same set of compounds hence share some overlaps that are presented in Venn diagrams for (**A**) non-antimicrobials and (**B**) antimicrobials.

**Figure 2 pharmaceuticals-13-00204-f002:**
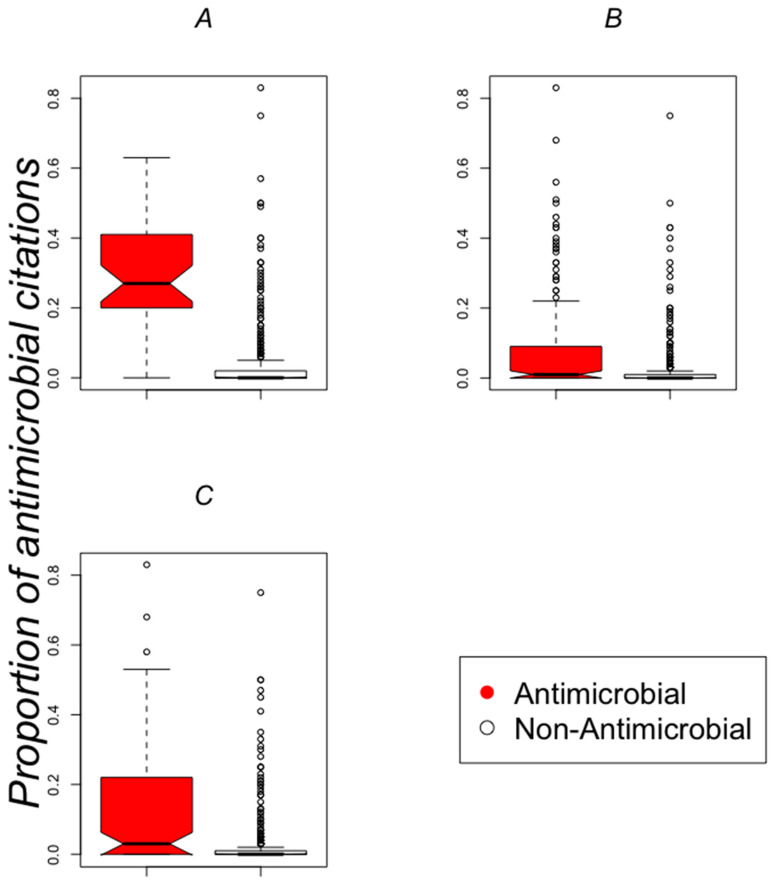
PubMed citations with ontological terms associated with antimicrobial activity. (**A**) Anti-infective, (**B**) anti-gut1 and (**C**) anti-gut4 data sets. The images present the distribution of the proportion of publications that included an ontological term related to antimicrobial activity with respect to the total number of publications reported for every compound tested by Maier and collaborators; to see the actual ontological terms, please refer to Table 1. The data are presented in box plots, where the first and third quartile are represented below and above the black line that corresponds with the mean value of the distribution. Every box plot presents the data for each of the 572 compounds that presented at least one publication with ontological terms associated with antimicrobial activity, out of the 1181 compounds tested by Maier and collaborators. Red boxes represent antimicrobials, white otherwise.

**Figure 3 pharmaceuticals-13-00204-f003:**
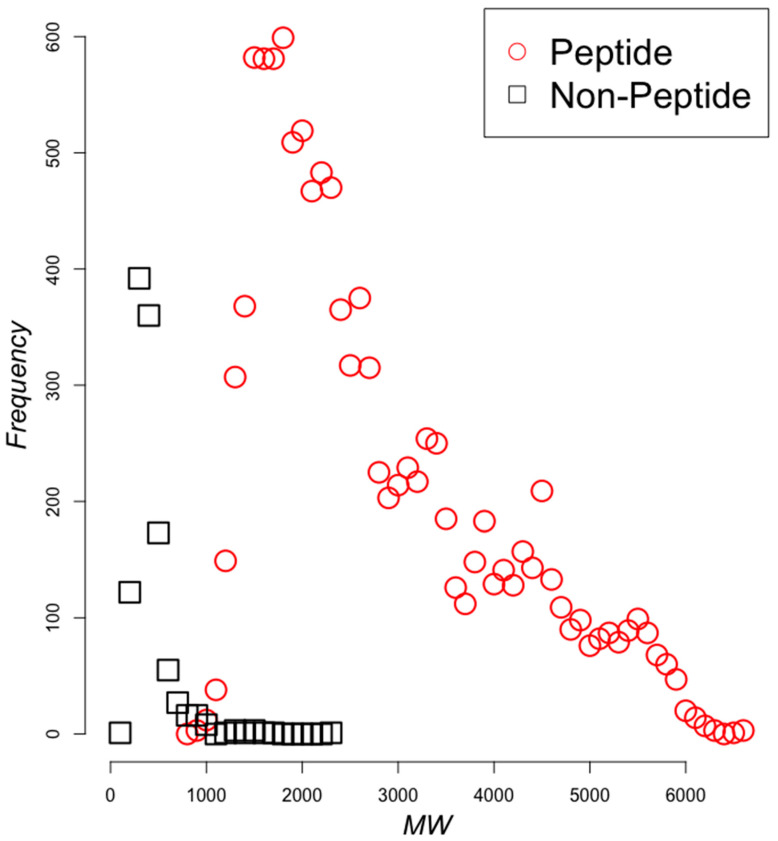
Molecular weights comparison between peptides and non-peptides in the training set. Red squares represent peptides and non-peptides are black squares. The y-axis presents the observed frequency of molecules within the specified range of molecular weights (x-axis). Molecular weights were accumulated in bins of size 100 Da (0–100, 101–200,…, 6401–6500, 6501–6600). Thus, every symbols correspond with the number of molecules observed every range of 100 Da in molecular weight.

**Figure 4 pharmaceuticals-13-00204-f004:**
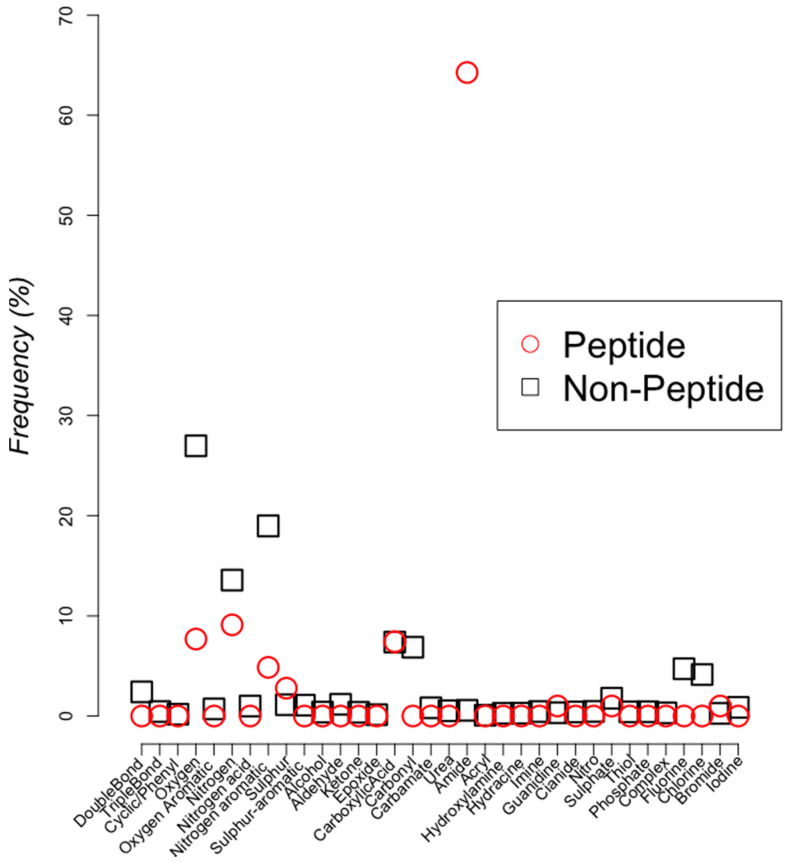
Chemical groups comparison. Functional group frequency (%) is compared between peptides (red circles) with non-peptidic chemical compounds (black squares). The names of the functional groups found in both sets are indicated on the x-axis (see Methods).

**Figure 5 pharmaceuticals-13-00204-f005:**
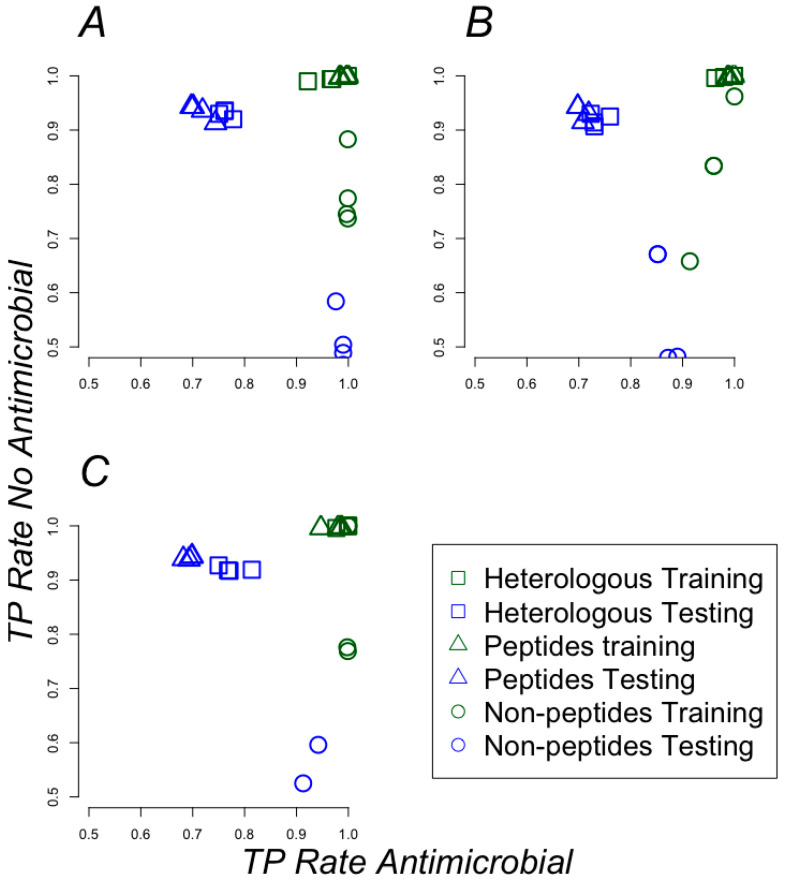
Learning efficiency based on physicochemical features. (**A**) Anti-infective, (**B**) anti-gut1, and (**C**) anti-gut4 training sets. True positive (TP) rates (number of correct predictions divided by the number of positive instances) for classifying antimicrobials and non-antimicrobials are shown for models trained with each training sets (green symbols) and for a 10-fold cross-validation test (blue symbols). Squared symbols correspond with models trained with heterologous data (peptides and non-peptide compounds); triangle symbols represent models trained with peptides; circular symbols represent models trained with non-peptidic compounds.

**Figure 6 pharmaceuticals-13-00204-f006:**
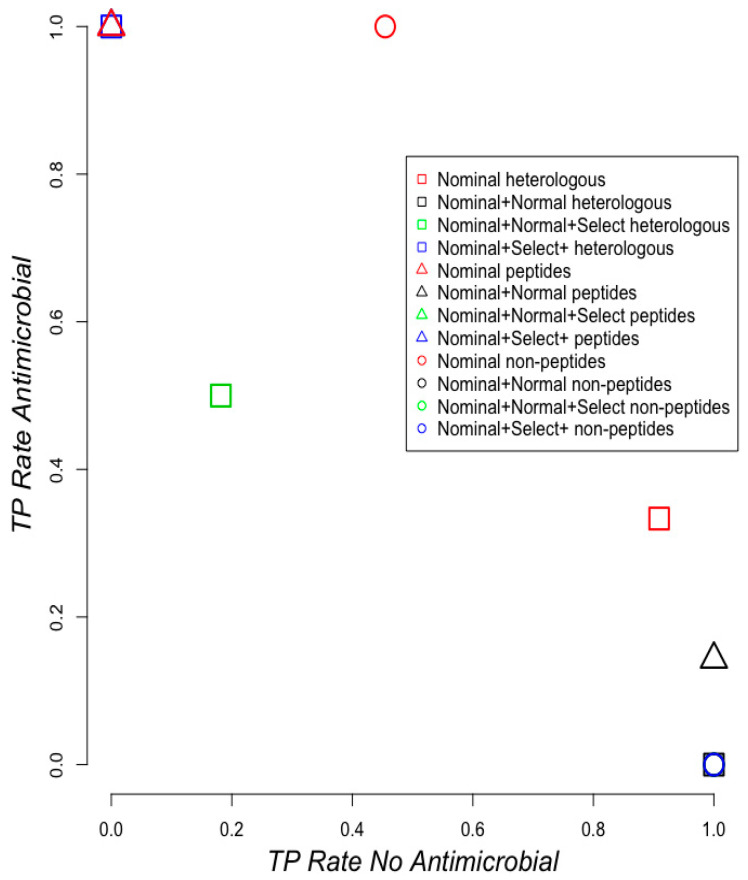
Testing models on human metabolites. True positive (TP) rates (number of correct predictions divided by the number of true positives) for classifying antimicrobials and non-antimicrobials are shown for models trained in the anti-gut4 dataset using heterologous data (peptides and non-peptide compounds; squares), only peptides (triangles) and non-peptidic compounds (circles). There are four instances of each symbol corresponding with the four data representation used: (i) nominal (red); (ii) nominal and normalized (black); (iii) nominal normalized and selected attributes (green); and (iv) nominal and selected attributes (blue) (see Methods).

**Figure 7 pharmaceuticals-13-00204-f007:**
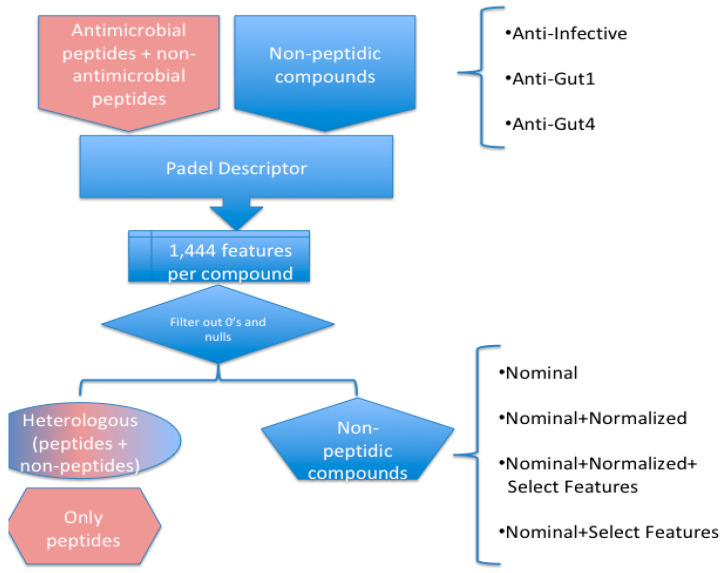
Dataset construction. The steps described to build the datasets (anti-infective, anti-gu1, and anti-gut4) used throughout this work are described. From 1,181 non-peptidic compounds and 11,545 peptides, three sets were derived: heterologous, only peptides and non-peptidic compounds. Each of these sets was converted into ARFF format using the specified combination of the following procedures: nominal representation of the class (antimicrobial or non-antimicrobial), features normalized, and/or feature selection.

**Table 1 pharmaceuticals-13-00204-t001:** Antimicrobial ontological terms.

Antifungal agents pharmacology	Antifungal agents administration and dosage	Antifungal agents administration and dosage therapeutic use	Antifungal agents chemical synthesis chemistry pharmacology
Antifungal agents therapeutic use	Drug resistance fungal	Fungi drug effects	Virus replication drug effects
Antiviral agents therapeutic use	Anti-bacterial agents analysis	Anti-bacterial agents pharmacology	Anti-bacterial agents therapeutic use
Drug resistance bacterial	Drug-resistance multiple bacterial	*Mycobacterium tuberculosis* drug effects	Anti-bacterial agents
Anti-bacterial agents administration and dosage adverse effects	Anti-bacterial agents administration and dosage pharmacology	Anti-bacterial agents administration and dosage therapeutic-use	Anti-bacterial agents adverse-effects
Anti-bacterial agents chemistry	Anti-bacterial agents pharmacology therapeutic-use	Anti-bacterial agents toxicity	Bacterial infections drug-therapy
DNA-bacterial genetics	Drug-resistance bacterial-genetics	Gram-negative-bacteria drug-effects	Gram-positive-bacteria drug-effects
*Helicobacter* infection drug-therapy microbiology	*Helicobacter pylori*	*Helicobacter pylori* drug-effects	*Mycobacterium avium* complex-drug-effects

**Table 2 pharmaceuticals-13-00204-t002:** Top 10 features for the best model (random forest) during training.

Attribute Name ^1^	Description
BCUTw-1h	Eigenvalue based descriptor noted for its utility in chemical diversity
AATS5m	Average Broto—Moreau autocorrelation—lag 5/weighted by mass
MIC5a	Modified information content index (neighborhood symmetry of 5-order)/ weighted by atoms
AATS6m	Average Broto—Moreau autocorrelation—lag 6/weighted by mass
AATS7m	Average Broto—Moreau autocorrelation—lag 7/weighted by mass
IC5	Information content index (neighborhood symmetry of 5-order)
MIC4	Modified information content index (neighborhood symmetry of 4-order)
AATSC0m	Average centered Broto—Moreau autocorrelation—lag 0/weighted by mass
topoDiameter	Topological diameter (maximum atom eccentricity)
AATS0m	Average Broto—Moreau autocorrelation—lag 0/weighted by mass

^1^ Attribute ranking was based on the information Gain Ranking filter implemented in Weka (see Methods).

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
