# Peer review of "Relevant Features of Polypharmacologic Human-Target Antimicrobials Discovered by Machine-Learning Techniques"

_pharmaceuticals, 2020, doi:10.3390/ph13090204_

Round 1

Reviewer 1 Report

The authors of the article, "Relevant features of polypharmacologic human-target antimicrobials discovered by machine-learning techniques." have done an interesting study.

There are a few issues to address before accepting the manuscript.

1) Line 46-47: "they reported to have .......class of antipsychotics" reframe the sentence for the readers' legibility.

2) Table-1: reformat the table in columns.

3) Figure-2: Is the scale identical in figure-2, A, B, and C? Also, provide the red and white data bar description in the inset.

4) Is there any literature available that describes the antimicrobial activity taking the molecular structure/features into account?

Author Response

1) Line 46-47: "they reported to have .......class of antipsychotics" reframe the sentence for the readers' legibility.

We changed the description of this idea and marked this change in red in the new version of our work, to facilitate the identification of the change.

2) Table-1: reformat the table in columns.

We appreciate the suggestion. We changed the format of the table as indicated by the reviewer.

3) Figure-2: Is the scale identical in figure-2, A, B, and C? Also, provide the red and white data bar description in the inset.

We change the figure according to the suggestion by the reviewer. Figure 2 is now figure 3 in the revised version of our work.

4) Is there any literature available that describes the antimicrobial activity taking the molecular structure/features into account?

Yes, there is; we appreciate the note. We are now citing some of those references in our work in the discussion section.

Reviewer 2 Report

Here, Nava Lara and colleagues explore use ML to explore the features of antimicrobial small molecule drugs that might also have activity directed toward the human host. Overall, I think an interesting thought exercise worthy of publication despite some sloppiness with the production of this work and no actual

Comments:

  • There needs to be a more thorough introduction of anti-gut1 and anti-gut2 nomenclature
  • Figure 1: ‘amti-gut1’ should be anti-gut1
  • Figure 1 would be better represented as a bar chart, pie chart, or venn diagram and not a ppt slide. Similarly, this typeface should be sans serif like the other figures for consistency and professionalism.
  • Table 1: every entry should be a new row for ease of reading
  • Figure 4 there are many errors:

                                       ‘Doble bond’ should be double bond

                                        ‘Phenil’ should be “phenyl”

                                        ‘acid’ should likely be carboxylic acid or carboxylate for clarity

                                        ‘hydroxilamine’ should be hydroxylamine

                                        ‘Nitrogen acid’ needs more clarity, what functional group is this?

                                        ‘imina’ should be imine

  • Figure 4, I have a hard time believing that only 10-20% of small molecule anti-infectives have O or N within their structures. By default, this needs clarifying as amides (70-80% in peptides [also should be 100% by default]) have N and O by their constitution. I think there are serious errors with this analysis that give the reviewer pause.
  • Figures all should have labels as easily visible within the figure. I had difficulty understanding these quickly as I had to read the legends and then draw on each figure to make sense of them. E.g., Figure 3 should have a black box on the figure itself labeled non-peptides and a red box labeled peptides.

Author Response

1) There needs to be a more thorough introduction of antigut1 and anti-gut2 nomenclature

We have added a more detailed description of Anti-Gut1 and Anti-Gut4 and we add a new figure that summarizes the steps to build these datasets (new Figure 1). The changes are marked in red to facilitate the location of the changes in the text.

2) Figure 1: ‘amti-gut1’ should be anti-gut1. Figure 1 would be better represented as a bar chart, pie chart, or venn diagram and not a ppt slide. Similarly, this typeface should be sans serif like the other figures for consistency and professionalism.

We are now presenting Figure 2 (before Figure 1) in two Venn diagrams and changed the legend to describe this new representation.

3) Table 1: every entry should be a new row for ease of reading

We have changed the presentation of data in Table 1 to address this suggestion.

4) Figure 4 there are many errors:

‘Doble bond’ should be double bond

‘Phenil’ should be “phenyl”

‘acid’ should likely be carboxylic acid or carboxylate for clarity

‘hydroxilamine’ should be hydroxylamine

‘Nitrogen acid’ needs more clarity, what functional group is this?

‘imina’ should be imine

We appreciate the note; we have corrected the names of these chemical groups in the figure. Nitrogen acid is detected as a functional group in SMILES containing [N+] and corresponds with nitrogen with positive charge as a consequence of a forth-coordinated bond, excluding nitro group.

5) Figure 4, I have a hard time believing that only 10-20% of small molecule anti-infectives have O or N within their structures. By default, this needs clarifying as amides (70- 80% in peptides [also should be 100% by default]) have N and O by their constitution. I think there are serious errors with this analysis that give the reviewer pause.

The reviewer is correct in that every peptide must contain an amide and that many more than 20% of small molecules include oxygen in their structures. However, we are reporting normalized percentages. Since the number of peptides and non-peptidic compounds differ, and we wanted to compare the functional groups between these two groups of compounds, we normalized by the total number of functional groups per group of compound. Thus, amides are 70-80% of all possible functional groups detected in all peptides, while oxygen or nitrogen represent each 10-20% of all functional groups in small molecules. To clarify this we changed the description about these results.

6) Figures all should have labels as easily visible within the figure. I had difficulty understanding these quickly as I had to read the legends and then draw on each figure to make sense of them. E.g., Figure 3 should have a black box on the figure itself labeled non-peptides and a red box labeled peptides.

We have added legends on all figures to attend this suggestion.

Reviewer 3 Report

While the premise of using machine learning to detect metabolites of human physiology which also have antibacterial activity is interesting, there are significant omissions of information needed for peer review and reproducibility. 

Minor

Line 31-32 “…been noted that although the mechanism of action for some drugs is well studied (e.g., GPCRs [4]⁠, 31 protein kinases [5]⁠ and penicillin [6]⁠)”

GPCRs and kinases are not drugs, but GPCR and kinase inhibitors are. This sentence needs to be modified so that the examples given (GPCRs, protein kinases and penicillin) are all either mechanism of action or drugs/drug classes

Line 53 “…peptide and non-peptide antimicrobials compounds…” should have antimicrobial rather than antimicrobials

Minor noun class agreements and plurality grammatical check is needed throughout. Line 146-148 has a clunky sentence which needs improvement.

Major

Beginning with Line 63 “We built three different training datasets: anti-infective, anti-gut1 and anti-gut4 (see Methods);”

I can find no mention of whether the Anti-Gut1 set is included in the AntiGut4 set. We are only told Anti-gut1 contains antimicrobials acting on 1 or more gut species while Anti-Gut4 contains antimicrobials acting on 4 or more gut species. Should the definition of the Anti-Gut1 set actually be antimicrobials acting on 1-3 gut species? This ambiguity needs to be resolved in the manuscript.

Additionally, this section has me very confused regarding the classification task and the composition of each dataset. The authors state that they use 1181 compounds from Maier, yet the total number of antimicrobials in the Anti-Infective, Anti-Gut1 and Anti-Gut4 sets only sums to 788 compounds. Also, the non-antimicrobials seemingly appear out of thin air. Where were these compounds acquired from? The authors need to establish the nature of each set’s assembly, sourcing and classification composition.

Line 130-131 “…for that end, we added 7999 antimicrobial peptides and 3546 non-antimicrobial peptides to each of these sets for the heterologous training set construction…”

From what informatics database were these structures sourced? This is essential information to be included in both the main text and the methods. Additionally, the rational for the selection of these specific structures needs to be included as well. This should be the case for all datasets assembled by the authors. The source and the reason for inclusion/exclusion if the whole source is not used need to be added for every set.

Line 146-147 “Indeed, it has been recently noted that no automatic procedure to accomplish this goal leading to the development of a machine-learning-based approach for that goal”

This statement needs citations to support its claims.

Line 152 The PaDel-descriptor needs a citation.

Line 157 The machine learning algorithm needs to be mentioned here. The authors later state that they used a random forest algorithm, but they do not give mention of the number of trees, how entropy in learning was calculated, and other core properties needed to reproduce this work.

Beginning Line 168 “12 different models were generated for each training set: 4 sets for heterologous training, 4 sets using only non-peptidic chemical compounds and other 4 sets using only peptidic chemical compounds”

How were each of these sets created? This needs to be included in this part of the paper.

Beginning line 190 “Please note that all these features are related to the information or graph theory parameters of chemical groups, rather than physicochemical attributes”

After spending the entirety of section 2.2 discussing physiochemical properties, the authors then inform the reader that they also used graphical methods to represent these structures. At no point in the manuscript until this mention here have they discussed the calculation of the 507 features used for learning. This is a major omission.

Author Response

Minor

1) Line 31-32 “…been noted that although the mechanism of action for some drugs is well studied (e.g., GPCRs [4] , 31 protein kinases [5] and penicillin [6] )” GPCRs and kinases are not drugs, but GPCR and kinase inhibitors are. This sentence needs to be modified so that the examples given (GPCRs, protein kinases and penicillin) are all either mechanism of action or drugs/drug classes

We appreciate the note; we changed the text to clarify this aspect (lines 36-37 in the new version). All changes are marked in red in the text.

2) Line 53 “…peptide and non-peptide antimicrobials compounds…” should have antimicrobial rather than antimicrobials

We have corrected this error.

3) Minor noun class agreements and plurality grammatical check is needed throughout. Line 146-148 has a clunky sentence which needs improvement.

We have checked the spelling and grammar of our work and corrected this particular problem.

Major

4) Beginning with Line 63 “We built three different training datasets: anti-infective, anti-gut1 and anti-gut4 (see Methods);”

I can find no mention of whether the Anti-Gut1 set is included in the AntiGut4 set. We are only told Anti-gut1 contains antimicrobials acting on 1 or more gut species while Anti-Gut4 contains antimicrobials acting on 4 or more gut species. Should the definition of the Anti-Gut1 set actually be antimicrobials acting on 1-3 gut species? This ambiguity needs to be resolved in the manuscript.

We have changed the description about the construction of the datasets to clarify this point. We include now Figure 1 to summarize the steps followed to build the datasets.

5) Additionally, this section has me very confused regarding the classification task and the composition of each dataset. The authors state that they use 1181 compounds from Maier, yet the total number of antimicrobials in the Anti-Infective, Anti-Gut1 and Anti-Gut4 sets only sums to 788 compounds. Also, the non-antimicrobials seemingly appear out of thin air. Where were these compounds acquired from? The authors need to establish the nature of each set’s assembly, sourcing and classification composition.

The dataset from Maier has 1,181 FDA-approved compounds. We have changed the description of the datasets to clarify this aspect of our work. We appreciate this note. We realized the labels in figure 1 were mixed: antimicrobials were labeled as non-antimicrobials. We have corrected this error.

6) Line 130-131 “…for that end, we added 7999 antimicrobial peptides and 3546 non-antimicrobial peptides to each of these sets for the heterologous training set construction…”

From what informatics database were these structures sourced? This is essential information to be included in both the main text and the methods. Additionally, the rational for the selection of these specific structures needs to be included as well. This should be the case for all datasets assembled by the authors. The source and the reason for inclusion/exclusion if the whole source is not used need to be added for every set.

We appreciate the note. We have changed the description of this part of our work. We realized that we have omitted unintentionally many details, however all the data to reproduce our work is available at Github (https://gdelrioifc.github.io/PolyHAM/).

7) Line 146-147 “Indeed, it has been recently noted that no automatic procedure to accomplish this goal leading to the development of a machine-learning-based approach for that goal”

This statement needs citations to support its claims.

The citation was included in the Methods section. We are now including the citation to the work by Ertl in the results section.  

8) Line 152 The PaDel-descriptor needs a citation.

The citation was in Methods, but we are including now this citation in the main text too.

9) Line 157 The machine learning algorithm needs to be mentioned here. The authors later state that they used a random forest algorithm, but they do not give mention of the number of trees, how entropy in learning was calculated, and other core properties needed to reproduce this work.

We did not use a specific machine-learning algorithm. Instead, we used an optimization algorithm that searched among the algorithms implemented in Weka the best one to model the data. In section 2.3 we stated: “It is worth to mention that AutoWeka includes the state-of-the-art machine-learning algorithms like SVM, Random Forest, and Logistic Regression among others, so the resulted learning model can be considered as the most suitable for the classification task at hand”. For every algorithm identified in this procedure, we report its parameters in Supplementary Files S43 (anti-infective), S44 (anti-gut1) and S45 (anti-gut4).

10) Beginning Line 168 “12 different models were generated for each training set: 4 sets for heterologous training, 4 sets using only non-peptidic chemical compounds and other 4 sets using only peptidic chemical compounds”

How were each of these sets created? This needs to be included in this part of the paper.

All the methodological details were presented in the Methods section. But we recognize that our description was not clear enough. We appreciate this note. We are including a more detailed description of the datasets both in Methods and in the Results section.

11) Beginning line 190 “Please note that all these features are related to the information or graph theory parameters of chemical groups, rather than physicochemical attributes”

After spending the entirety of section 2.2 discussing physiochemical properties, the authors then inform the reader that they also used graphical methods to represent these structures. At no point in the manuscript until this mention here have they discussed the calculation of the 507 features used for learning. This is a major omission

We appreciate the note. The version of Padel Descriptor used allowed for the calculation of up to 1,444 2D features. After eliminating features that were equal to zero or null in half or more of the instances (this procedure was conducted to all data sets as indicated in Methods), we obtained 507 features in the case of the best model; in the description of the best model, we specified that this model used heterologous data with nominal classes and no attribute selection was required. We are now including a more detailed description of the datasets used to address this limitation and a new Figure (figure 1) that summarizes the procedures to build these sets.

Round 2

Reviewer 3 Report

The authors have addressed previous concerns about transparency. However, minor punctuation errors (placement of spaces in text) remain throughout and need to be addressed before publication.

See below for some examples:

"478 monocytogenes in combination with a natural antimicrobial peptide [45]⁠ . In this case,..."

"429 body and only act on a specific target [33]⁠ . However, the..."

"432 in microbes because they act on multiple targets [34]⁠ . Thus..."

Author Response

1) The authors have addressed previous concerns about transparency. However, minor punctuation errors (placement of spaces in text) remain throughout and need to be addressed before publication.

See below for some examples:

"478 monocytogenes in combination with a natural antimicrobial peptide [45]⁠ . In this case,..."

"429 body and only act on a specific target [33]⁠ . However, the..."

"432 in microbes because they act on multiple targets [34]⁠ . Thus..."

We appreciate the note. These problems are the consequence of the reference manager we are using. These spaces showed up only when we use openoffice, so we eliminated those spaces using that software. We then realized that some of us did not see the spaces after deleting them, but some of us noticed the spaces in the PDF. Thus, we believe this is a problem that arises from using different operating systems. We hope the new version is more reliable in the format than the previous one.